# Paper Thin? The Evidence for 12th-Century Gothic Design Drawings

Robert Bork

School of Art and Art History, The University of Iowa, Iowa City, IA 52242, USA; robert-bork@uiowa.edu

**Abstract:** No Gothic design drawings on paper or parchment have survived from the 12th century, and only a few have survived from the 13th century. For this reason, most recent scholars tend to concur at least broadly with Robert Branner, who argued in an influential 1963 article that such drawings were first produced only after 1200. This conclusion deserves critical re-examination, however, for two principal reasons. First, the continuity of the Gothic architectural tradition in both time and space strongly suggests that early Gothic builders used similar techniques to those used by their late Gothic successors. From this perspective, the lack of surviving design drawings from before 1200 seems likely to reflect their disappearance over time, rather than their not being used in the crucial period when the conventions of Gothic design and construction were first coming into focus. A second argument for the use of drawings in the 12th century comes from consideration of early Gothic buildings, whose complex and carefully calibrated forms would be literally inconceivable without such graphic aids. Churches such as Saint-Denis Abbey and Notre-Dame in Paris, for example, already display a level of geometrical sophistication and coherence that argues strongly for the use of scaled drawings in their original conception.

**Keywords:** drawings; Gothic architecture; Robert Branner; Notre-Dame; Paris; Saint-Denis Abbey; parchment; geometry; twelfth century; plan; elevation

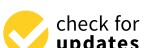



## 1. Introduction

How pervasive was the use of drawings in the Gothic architectural design process? This seemingly straightforward question has proven surprisingly difficult to resolve. Hundreds of Gothic design drawings on paper and parchment have survived, but most of these are from the Germanic world, and nearly all of them postdate 1300. By contrast, only a few have survived from the 13th century, and none have survived from the 12th century. For this reason, most recent scholars tend to concur at least broadly with Robert Branner, who argued in an influential 1963 article that such drawings were first produced only after 1200 (Branner 1963). This conclusion deserves critical re-examination, however, for two principal reasons, above and beyond the simple fact that the absence of evidence should not be misunderstood as evidence of absence. First, Branner's argument raises important questions about the continuity and coherence of the Gothic tradition itself, because it suggests that design drawings were not being used in the crucial period when the conventions of Gothic design and construction were first coming into focus, even though they were clearly used in many later Gothic contexts. A second and more powerful argument for the use of drawings in the 12th century comes from consideration of early Gothic buildings, whose complex and carefully calibrated forms would be literally inconceivable without such graphic aids. The present essay will address these issues by briefly reviewing the evidence that Branner and subsequent scholars have considered, by outlining what might be expected of designers working with and without drawings, and by showing that twelfth-century churches such as Saint-Denis Abbey and Notre-Dame in Paris already display a level of geometrical sophistication and coherence that strongly suggests the use of scaled drawings in their original conception.

## 2. Working Backwards from the Known

In attempting to assess the role of drawing in early Gothic design, it can be helpful to begin by working backwards from the later and at least somewhat better-documented phases of the Gothic era. Roughly 650 Gothic drawings on paper and parchment are known today, of which more than 400 are in Vienna (Böker 2005). This strikingly skewed distribution reflects the fact that only Vienna, of all European cities, preserves a substantial portion of the original medieval drawing collection associated with its principal local church. The workshop of this church, the Stephansdom, was certainly important in the late Middle Ages. Indeed, it was recognized at a masonic meeting in 1459 as one of the four supreme lodges of the Holy Roman Empire, along with those of Bern, Cologne, and Strasbourg (Brehm 2013). The three latter workshops would have had comparable drawing collections in the late Gothic era, but only small fractions of them have survived: one major drawing at Bern, six at Cologne, and twenty-eight at Strasbourg. Similar large collections would once have been found at other major centers across Europe where only a few parchment drawings now remain, such as Milan, Paris, Salamanca, Seville, and Ulm. Further evidence for the ubiquity of such drawings in the late Gothic era comes not only from smaller centers where examples still survive, such as Leuven, Orvieto, and Regensburg, but also from written documents attesting to their use.[1] The unique size of the Vienna collection, therefore, surely reflects the vagaries of uneven preservation, rather than its uniqueness in the Gothic era itself.

Despite its current singularity, therefore, the Vienna collection can offer a valuable perspective on the mixtures of drawing types that would have been produced in a typical late Gothic workshop. Nearly all of the Viennese drawings are either ground plans or elevations, rather than 3D views, and nearly all were carefully executed with compass and straightedge, although gargoyles, crockets, and similar sculpted details were often drawn freehand. These medieval "blueprints" thus display a high degree of geometrical order, which can be intuited by the casual viewer and confirmed by proportional analysis (Bork 2011). Some of the parchment drawings are large and impressive, with the elevation drawings for the Stephansdom's planned north tower, in particular, reaching heights of well over four meters; such charismatic drawings were likely used to illustrate alternative ideas for the project patrons, as well as for the architects themselves. Most of the parchment drawings, however, are smaller, and others executed on paper, sometimes as copies of parchment originals, should perhaps be understood as student exercises.[2] Nearly all of the drawings can be related to specific building projects, but they often show schemes slightly different to what was actually built, suggesting that they were used in the planning process, rather than depicting previously finished structures. And while most of the drawings now in Vienna depict components of the Stephansdom, many others illustrate other earlier monuments that may have inspired its builders, such as the Prague Cathedral, or later ones designed by members of its workshop, such as Vienna's fifteenth-century city hall. These patterns of relationship underscore the important role that drawings served in the Gothic period as vectors for the transfer of architectural information between sites and between builders. This, in turn, helps to explain how the most prestigious late medieval builders could simultaneously manage multiple projects in different cities. In terms of dating, finally, most of the Viennese drawings appear to date from the fifteenth or very early sixteenth century, but some of them, notably including those depicting the Prague Cathedral, likely date back to the fourteenth century.

Gothic drawings from the thirteenth century are far rarer than those from the later Middle Ages, but at least three sets of them have survived to achieve prominence in the scholarly literature. The most recent, and most immediately comprehensible, depict alternative designs for the façade of the Strasbourg Cathedral (Bork 2011, pp. 62–96). The largest of them, known as Strasbourg Plan B, shows the full vertical extent of a façade broadly similar, at least in its lower reaches, to the one constructed in the beginning of 1277. Strasbourg Plan A, which was presumably drawn several years earlier, shows a simpler scheme similar in its articulation to the transept frontals of Notre-Dame in Paris, which

were planned from around 1250. Both drawings have rigorous geometrical order. Plan A, for example, was organized around a scheme of interlocking of squares and equilateral triangles, as Figure 1 shows.

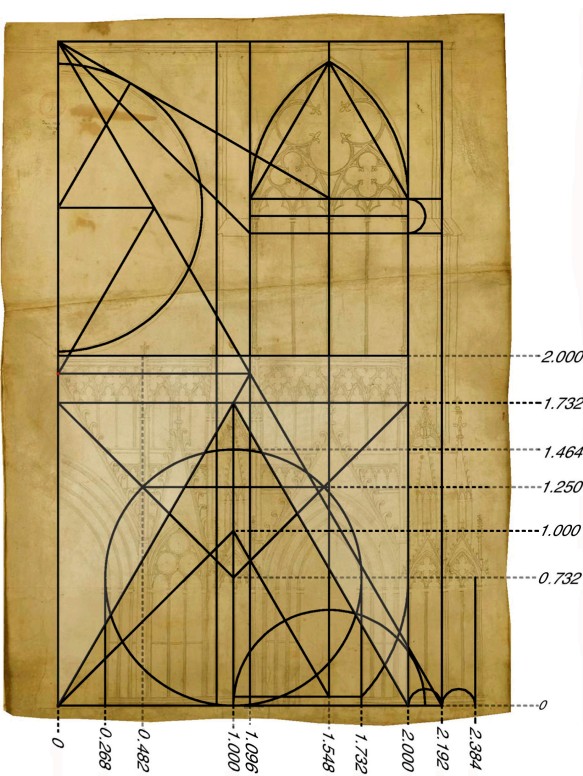

**Figure 1.** Strasbourg Plan A, geometry [Strasbourg, Musée de l'Œuvre Notre-Dame, Inv. OND 1, with geometrical overlay by the author].

Meanwhile, in terms of facture, Strasbourg Plans A and B were drawn with high precision, and in the same basic manner as the Viennese drawings made two centuries later, which clearly belong to the same genre of architectural workshop drawings. So, although Strasbourg Plan A likely ranks as the oldest surviving example of this type, it seems to represent an already mature tradition, rather than an early experiment.

A second set of thirteenth-century drawings comes from the so-called Reims Palimpsest (Branner 1958; Murray 1978; Bork 2011, pp. 42–54). They survived, however, only as faint traces, because they were erased already by 1270, when the sheets they were drawn on were re-used to record an obituary for the Reims Cathedral Chapter. These traces include depictions of elements such as choirstall components, tracery, finials, and two façade schemes, which seem on stylistic grounds to date from around 1230 and 1250, respectively. The facades cannot, however, be directly associated with any specific building project, and they may actually represent designs for microarchitectural shrines (Kurmann 2002). Such ambiguities notwithstanding, the drawings from the Reims Palimpsest appear to have been drawn carefully and with geometrical precision, like the slightly later façade drawings from Strasbourg, and the considerably later drawings from centers such as Vienna.

The third and most famous set of thirteenth-century drawings comes from the manuscript known as the Portfolio of Villard de Honnecourt, illustrating not only architectural subjects but also machines, animals, and humans, sometimes organized around geometrical armatures. The drawings in the portfolio, usually dated to circa 1220–1230, differ from those mentioned previously, not only in their wider range of subject matter but also in their looser execution, which makes them seem more like sketches than like blueprints. So, while some scholars from the nineteenth century to the present day have assumed that Villard de Honnecourt was an architect, many others have argued that the itinerant draftsman responsible for the drawings was a carpenter, a goldsmith, or a cleric.[3] It seems clear,

regardless, that Villard had first-hand access to the cathedral workshop at Reims, since he carefully depicted details such as the way that individual stones would be assembled into columns, which a more casual visitor to the site would have missed.

Some of his other drawings depict the eastern chapels and the main elevation of Reims Cathedral (Figure 2) from both the inside and the outside, introducing useful complementary points of view, while the chapel drawings incorporate curved baselines that hint at their convex and concave profiles. Villard also included pseudo-perspectival depth cues in his depiction of the Laon Cathedral tower elevation and in many of his representations of machines, but most of his architectural drawings are plans or elevations conceptually equivalent to the later and larger workshop drawings surviving in collections such as Vienna's (Bork 2011, pp. 31–42). So, while Villard's precise relationship to the architectural profession remains obscure, the survival of his portfolio certainly hints at the existence of a well-established culture of architectural drawing by the second quarter of the thirteenth century.

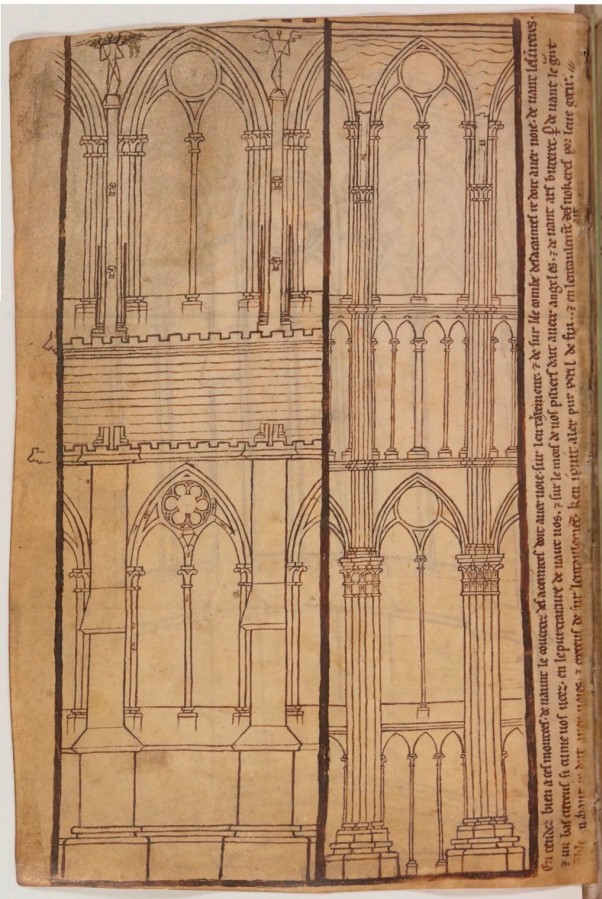

**Figure 2.** Portfolio of Villard de Honnecourt, interior and exterior elevations of the straight bays of Reims Cathedral, fol. 31v, Paris, Bibliothèque nationale de France, MS fr. 19093 [Bibliothèque nationale de France].

Although it seems highly unlikely that Villard's coincidentally preserved drawings were the first of their kind to be produced, they appear to be the oldest to survive, since most of the earlier architecturally related drawings belong to one of two other, very different genres: on the one hand, drawings of details incised in stone or plaster, often at full scale; on the other, diagrammatic illustrations of theological concepts in manuscripts. Drawings of the former type evidently had practical function for builders on the worksite, likely for working out the form of specific elements, or even for establishing templates used for stone cutting.[4] Surviving examples of the latter type, although far removed from the architectural workshop, suggest that conventions of representation in plan and elevation were already

being employed in the twelfth century, as in the illustrations for Richard of Saint Victor's *In visionem Ezechielis* (Kinsella 2023).

Two other rare examples of drawing on parchment from before 1200 also deserve mention in this context, both showing large monastic churches in their architectural contexts. The so-called Waterworks drawing bound into the Eadwine Psalter in the late twelfth century, for example, shows a view of Canterbury's Romanesque cathedral and its outbuildings, including cloisters, fountain houses, and the pipes connecting them (Fergusson 2016). Despite this documentary character, the architectural drawings here are rather impressionistic, recalling representations of buildings in traditional manuscript illumination. The famous ideal plan of Saint Gall monastery produced in the early ninth century, conversely, appears more geometrically rigorous but more schematic, showing only the outlines of the building plans in question, with no architectural detail or information about the third dimension (Figure 3) (Horn and Born 1979; see also Büker 2020).

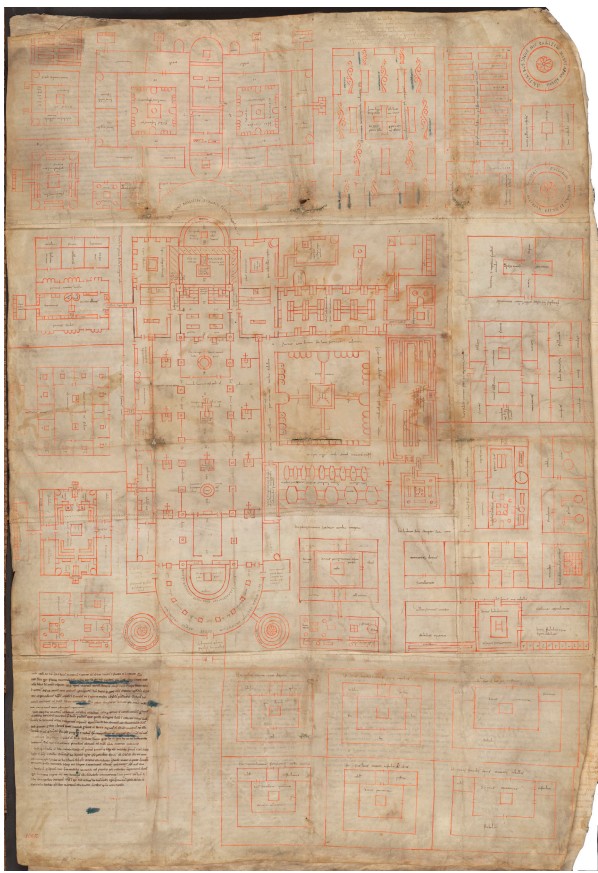

**Figure 3.** Saint Gall Plan, St. Gallen, Stiftsbibliothek, Codex Sangallensis 1092. [Wikimedia Commons].

## 3. The Debate about 12th-Century Drawings and Its Stakes

Confronted by this pattern of evidence, Robert Branner concluded that precise architectural design drawings on parchment emerged only from the thirteenth century onwards, and that Villard de Honnecourt was a witness to this process. Branner argued in his 1963 article that such drawings would not have been necessary earlier because "the technique of construction was still relatively simple and in all likelihood it was the patron who informed the 'architect' how he wanted the finished monument to look. Even if buildings such as Hildesheim, Cluny, and Fontenay were designed on arithmetical, or more likely, on geometrical principles, no matter how subtle or complex, the schemes could be worked out and the building erected from them without the intermediary of drawings... Stakes, cords, and simple instruments served in the layout of the plan, measuring rods were probably used for the elevation, and simple 'rules of thumb' based on long masonic

experience provided solutions to whatever stereometrical problems may have arisen". (Branner 1963, pp. 129–30).

Branner's rather breezy dismissal of drawings before Villard's era deserves critical scrutiny from several angles. First, while it might be true in principle that complex schemes could be implemented at full scale without the use of drawings, such an approach would be inherently risky and potentially problematic. Surely it would be easier for a designer to work out his ideas on parchment, where brainstorming could take place quickly and where errors could be fixed with a few scrapes of the knife, rather than at the worksite, where any such adjustments would be laborious, expensive, and perhaps even dangerous. Second, there is no technical reason to doubt that such drawings could have been made in the twelfth century. Compasses and straightedges were known and used in the early Middle Ages, as numerous manuscript compositions attest, and these tools could have been applied to architectural planning before 1200 just as easily as afterwards. Third, Branner was applying his argument not just to early medieval monuments, which often were fairly simple, but to all early Gothic buildings of the twelfth century, and even to the first generation of High Gothic cathedrals, such as Bourges, Chartres, and Reims, which were considerably more complicated. The idea that such buildings were simply improvised at the worksite, as Branner suggested, strains credulity.

Branner identified what he perceived as flaws in the oldest surviving Gothic drawings, including Villard's, arguing on this basis that drawing-based design practice was still in its awkward infancy in the mid-thirteenth century.[5] Here again, however, Branner's arguments seem uncharacteristically shaky. Since so little can be said conclusively about Villard's life and professional identity, for instance, flaws in his work might well reflect his status as an enthused amateur rather than the limits of the period's training for architectural professionals. And while Branner conceded that Villard must have seen extant workshop drawings at Cambrai Cathedral, he explicitly rejected the idea that he might have also done so at Reims, even though the copying of outmoded design drawings could help to explain many of the divergences between Villard's drawing and Reims Cathedral as built (Branner 1963, pp. 135–36). Geometrical analysis strongly suggests, for example, that the cathedral was originally designed with a total interior height equaling twice the arcade height, just as Villard's drawing shows (Bork 2022a, Figures 16 and 17). Moreover, while Villard's drawings do admittedly include some errors and glitches that cannot be fully explained away in such terms, the same was true of many later medieval drawings, as well, even including some of the late Gothic drawings from Vienna. The evidence of Villard's portfolio, therefore, seems fully compatible with the idea that there was already a robust tradition of Gothic architectural drawing around 1200, or even beforehand.

Although Branner's arguments against the use of parchment design drawings before Villard's day are less than compelling, his views have proven to be immensely influential. Most subsequent authors have tended at least tacitly to accept his point of view, while dismissing the possibility that such drawings were already broadly used in the twelfth century.[6] This trend has affected not only the specialized literature on Gothic architectural drawing, to which there have been many important contributions in recent decades, but also the broader scholarly conversation about Gothic design and building practice. In this context, particular attention should be given to the work of John James, who has argued since the 1970s that Chartres Cathedral and all previous Gothic churches were built by itinerant teams of masons working ad hoc season after season, rather than having the project controlled by a single designer or master plan.[7] James's thesis has proven controversial, and it has been rebutted by many experts.[8] Nevertheless, the case against it has been weakened by the unwillingness of most specialists to take seriously the idea that design drawings were widely used in the twelfth century. In one recent study, for example, Günther Binding invokes documentary descriptions of builders translating their schemes directly from their minds onto the building site to argue against the use of design drawings prior to Villard's era, even though he admits that the former procedure seems surprising on its face for the construction of great churches (Binding 2014, pp. 36–43).[9]

This debate has high stakes, since it bears directly on the question of whether Gothic architecture was a coherent historical phenomenon. If design drawings were not used until Villard's day, then the Gothic tradition would be best understood as having two very different regimes: a century of rapid innovation, during which builders working at full scale developed radical new forms such as the flying buttress and the spire, followed by three centuries during which designers worked in a wholly different mode using scale drawings, often emphasizing decorative detailing more than structural experimentation. If design drawings were used from the early twelfth century onwards, conversely, the same basic design practices would have been used throughout the Gothic era, and the radical formal and technical innovation of its early phase would be explicable as fruits of the same basic professional culture that produced the refinements and elaborations seen in its better-documented late phases. Of these two options, the latter appears more plausible, as one can see by considering both the overall pattern of Gothic architectural development in the twelfth century and the formal qualities of specific buildings designed in this period, such as Saint-Denis Abbey and Notre-Dame in Paris.

The use of drawings might be expected, in the abstract, to impact architectural culture in several distinct ways. For instance, the use of drawings should facilitate the development of buildings with complex designs which could not easily be conceived otherwise. More specifically, one would expect their complexity to be expressible in the two-dimensional form of plan and elevation drawings, rather than being based on simple volumes such as spheres and cubes. Structures built from drawings should also embody a high degree of geometrical order, with precise control of proportions in both the horizontal and vertical dimensions. One would further expect a drawing-based architectural culture to emphasize linear articulation and forms based on lines, especially in the plane. The use of drawings should also facilitate the exchange of architectural ideas between workshops separated from each other both in space and in time (Scheller 1995, pp. 181–83; See also Hillson 2020). All of these telltale characteristics can already be seen in the early Gothic architectural culture of twelfth-century France.

## 4. The Evidence from Saint-Denis

To demonstrate how well even the earliest Gothic buildings embody these principles, one can begin by considering the façade block of Saint-Denis, the first part of the abbey church rebuilt by Abbot Suger, which was consecrated in 1140. On the interior, the structure features rib vaults, newly fashionable elements that emphasize linearity in a way that the previously typical groin vaults did not; the ribs must have been drawn at full scale before their stones were cut and installed, and planning their form with small-scale drawings would have made sense as well. The complex forms of the massive compound piers supporting the vaults would also have been planned most readily in drawings (Figure 4).

In terms of proportion, the façade block is organized around a series of rectangles unfolded from the notionally square plans of the tower bases (Figure 5, bottom).[10] Although this fairly simple operation could have been conducted at full scale with minimal planning beforehand, the same cannot be said for the control of the vertical dimension. As the top half of Figure 5 shows, each portal fits neatly into a perfect square, while each of the three surrounding façade sections has a height:width ratio of 1:1.732, i.e., 1:$\sqrt{3}$, determined by the 60-degree slope of an equilateral triangle. The center of the rose window, meanwhile, lies exactly halfway up another set of equilateral triangles framed by the buttress axes and resting on the main horizontal molding over the façade's central window. These carefully calibrated relationships, which anticipate those seen in Strasbourg Plan A, were clearly planned with great care ahead of time, likely with scale drawings.

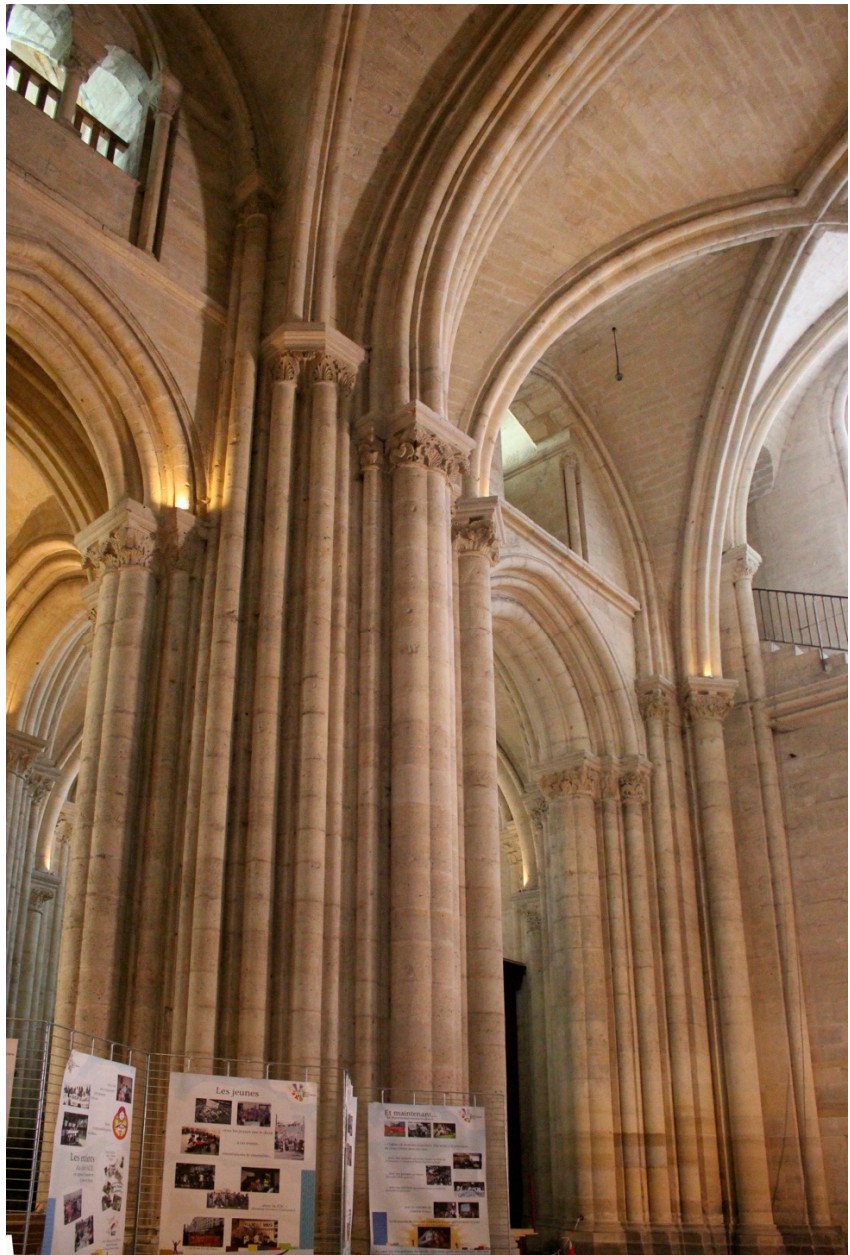

**Figure 4.** Saint-Denis Abbey, west façade block, interior [photo by the author].

The east end of Saint-Denis, consecrated in 1144, attests even more clearly than Suger's façade to the sophistication of geometrical planning in use at the dawn of the Gothic era. Its surviving twelfth-century components comprise a crypt and a main story, each equipped with seven radiating chapels (Figure 6).[11]

As Figure 7 shows, the radial structure of the crypt was carefully divided into seven wedges measuring 27 degrees each, which is the difference between the 45-degree angle found by bisecting a square and the 18-degree angle found by bisecting the 36-degree tip of a pentagonal star.[12] Each wedge corresponds to a chapel. The four chapels at the base of the composition all have the same size, which was found by inscribing the largest possible squares within those wedges and circles around those squares, as shown at the top of the figure. The next two chapels are slightly larger, being displaced outwards by dimensions set by the 36-degree triangles shaded green in the bottom half of the figure. The axial chapel is the largest of all, being further displaced in accord with another line sloped at 18 degrees to the composition's baseline. These brief sentences only begin to describe the basic geometry

of the crypt level, which also incorporates proportions based on the equilateral triangle to set the relative radii of its apse and ambulatory. These carefully calibrated geometrical relationships, and the even more complex patterns seen in the main level of the choir, likely would have been literally inconceivable without the use of scaled drawings, Branner's blithe dismissal of the complexity problem notwithstanding. Like the façade, moreover, the east end of Saint-Denis appears to have been meticulously designed in elevation as well as in plan, using a proportional scheme based on the stacking of equilateral triangles. This precise calibration of the vertical dimension implies the careful control of the worksite in accord with a coherent master plan over a period of at least several years, which again suggests the use of design drawings, while contradicting the model ad hoc year-by-year improvisation proposed by James.

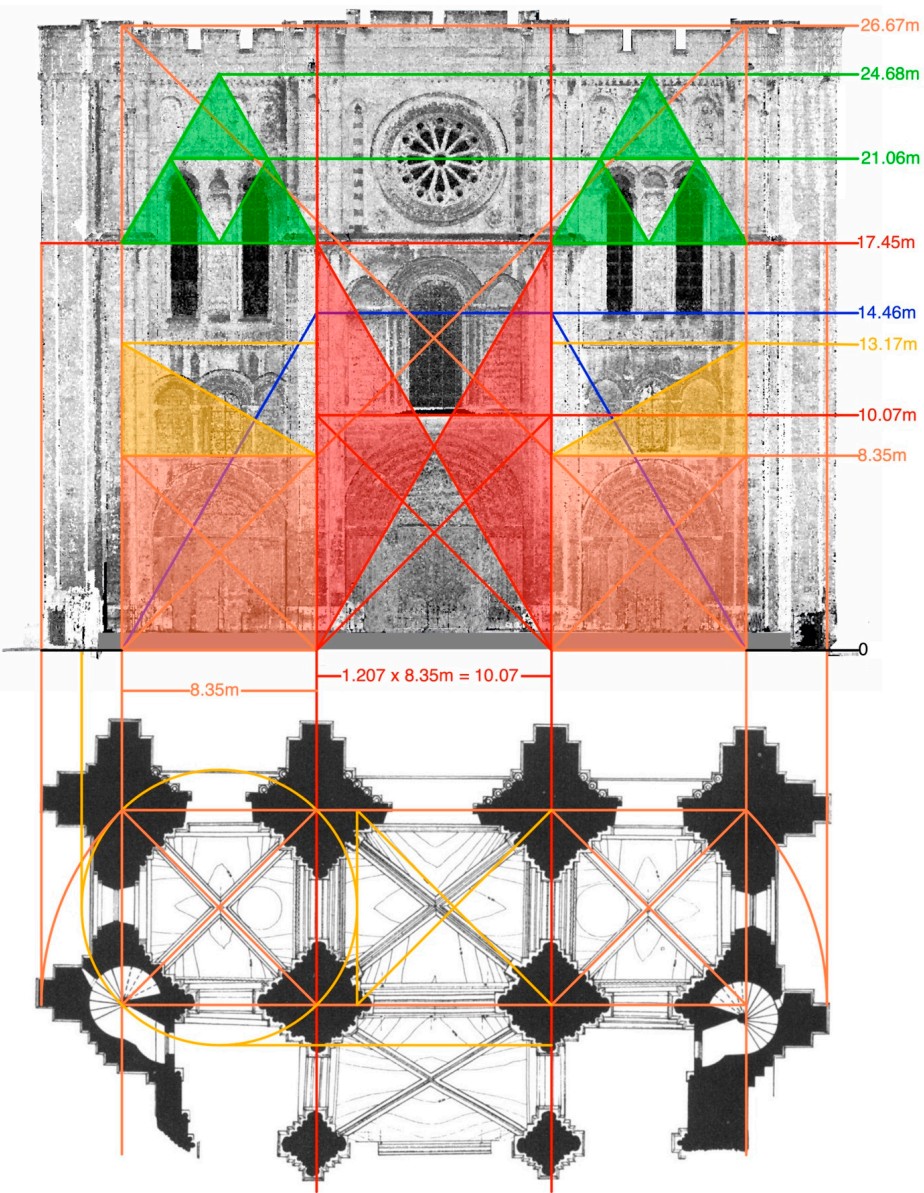

**Figure 5.** Saint-Denis Abbey, geometry of west façade laser scan data by Andrew Tallon, with geometrical overlays and annotation by the author].

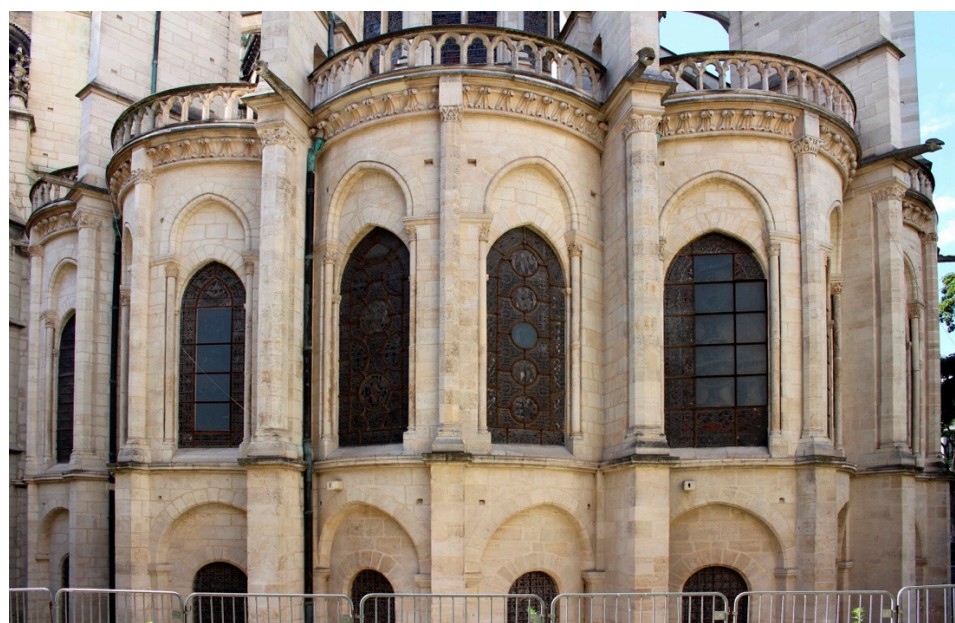

**Figure 6.** Saint-Denis Abbey, east end, showing crypt and radiating chapels [photo by the author].

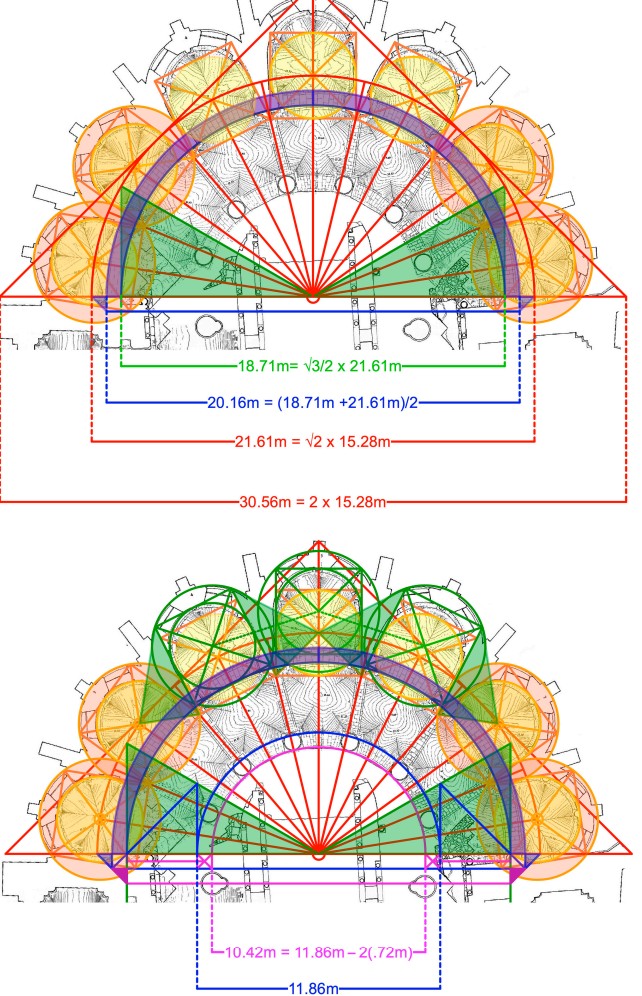

**Figure 7.** Saint-Denis Abbey, geometry of crypt, stages 1 (above) and 2 (below) [graphic by the author, based on photogram provided by Richard Nash Gould].

The reception of Saint-Denis also lends further qualified support to the idea that scaled drawings were used already in the twelfth century. Although no direct copies or clones of Saint-Denis were built in its wake, the decades after its completion saw the construction of several churches with similar but simpler east ends, featuring five chapels instead of seven, and featuring only a single ambulatory, as in the Saint-Denis crypt, rather than the double ambulatory seen in Suger's choir. These churches, which include Saint-Germain des Prés in Paris, Saint-Maclou in Pontoise, Saint-Leu in Saint-Leu d'Esserent, the Madeleine in Vézelay, and the cathedrals of Senlis and Noyon, all differ from each other in significant respects, but their plans have enough in common to suggest that drawings may have been used to share information among their workshops. The case of Vézelay is particularly interesting in this context because of its location in Burgundy, fairly far afield from Saint-Denis. The choir of Saint-Etienne at Caen, despite its location in Normandy, follows some aspects of the Saint-Denis plan even more closely; it features seven chapels, a double ambulatory, and unusually prominent buttresses at the base of the chevet. In terms of detailing and articulation, though, it differs significantly from Suger's choir. This disjunction could have arisen if, for example, the Norman builders had access to a schematic ground plan of Saint-Denis but not to a comparable elevation or detail drawing.[13] Further striking echoes of the Saint-Denis plan can be seen in the Cistercian abbey of Altenberg, Germany.[14] Although construction at Altenberg did not begin until 1259, more than a century after Suger's death, its builders carefully reproduced one particularly unusual feature of the Saint-Denis plan, namely the way that all the chapel windows are fully visible from the high altar, despite the fact that the altar is offset to the west of the chevet's geometrical center. The chevets of Altenberg and Saint-Denis are also precisely the same size, measured across the prominent buttresses at their chevet bases. These telling similarities strongly suggest that the designer of the Altenberg choir knew Saint-Denis from drawings, which would have revealed these subtleties in ways that a simple visit to the site would not. Such drawings, admittedly, could have been produced in the thirteenth century rather than in Suger's day, but it is tempting to imagine that original design drawings may have been kept on file at Saint-Denis, as they later would be at centers such as Strasbourg, Ulm, and Vienna.

## 5. The Evidence from Notre-Dame

The early history of Notre-Dame in Paris provides further indirect but strong evidence for the use of scaled design drawings already in the twelfth century. The choir of Notre-Dame, like that of Saint-Denis, has an elegant but subtle plan that could have been worked out far more readily at the drafting table than it could have been at the worksite (Bork 2022b). At Notre-Dame, the relative widths of the main choir vessel and the aisles are set by the proportions of the equilateral triangle, as shown schematically in Figure 8.

As the red lines in the bottom of the figure show, the entire radial portion of the chevet was built with a slight misalignment, being rotated 1.4 degrees to the north; this error seems to have been induced by sighting to the wrong intersection point in the geometrical scheme governing the composition, which is precisely the sort of error that one might expect if a scheme recorded in a drawing were translated incorrectly to the worksite. Despite this error, the choir's ground plan agrees well with the triangle-based scheme, as Figure 9 shows.

The same triangular geometry that governs the plan of the choir also governs its cross section, although with slight permutations. Further to the west, the plan of the nave and crossing together fit into a $\sqrt{3}$ rectangle framing two of the same large equilateral triangles that governed the choir plan, as shown in Figure 10. This striking result strongly suggests that the first designer of Notre-Dame established a master plan for the whole cathedral, even though it was completed by successors who modified its details.

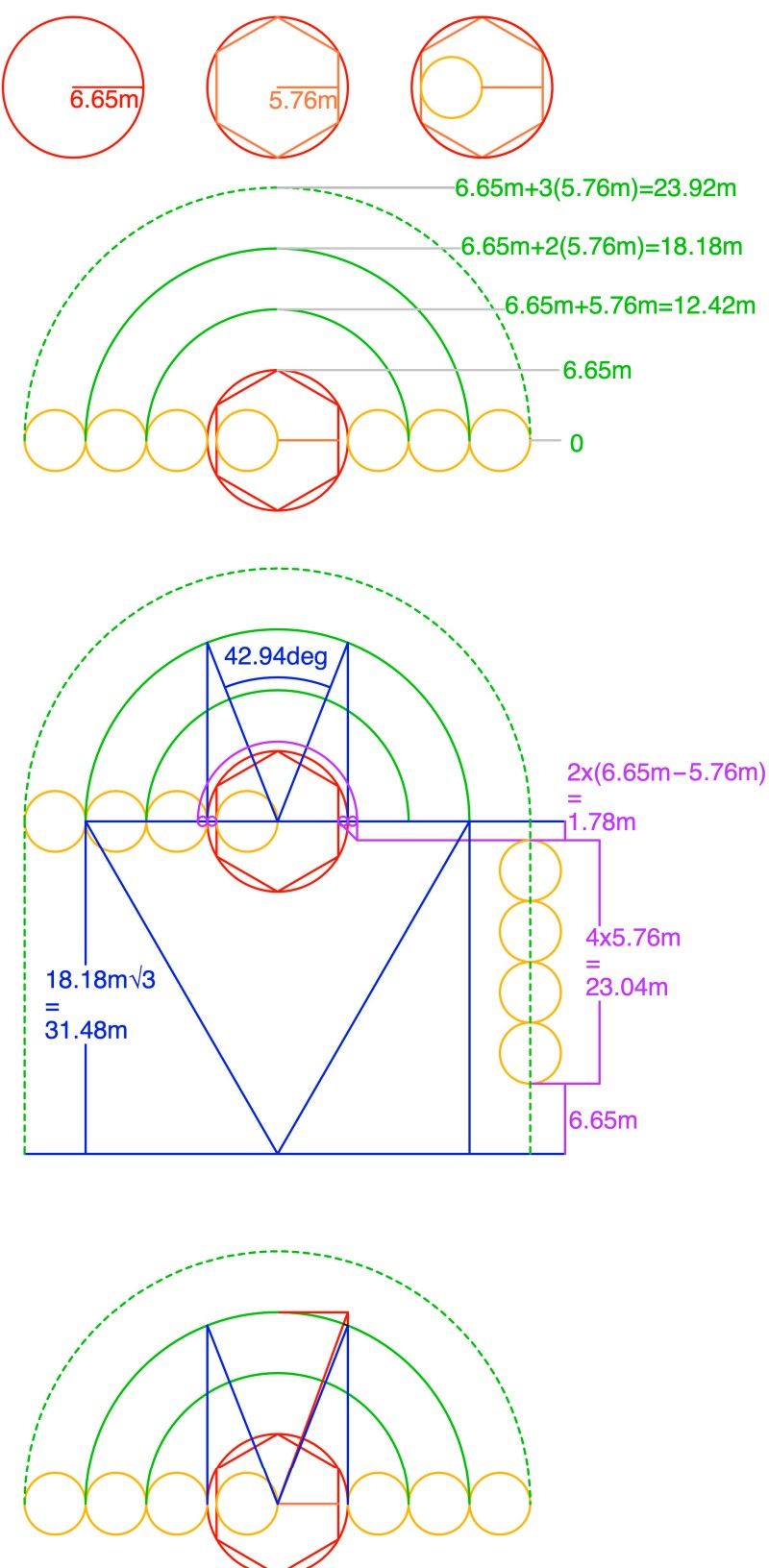

**Figure 8.** Schematic of Notre-Dame, Paris, showing steps in plan development of choir [graphic by the author].

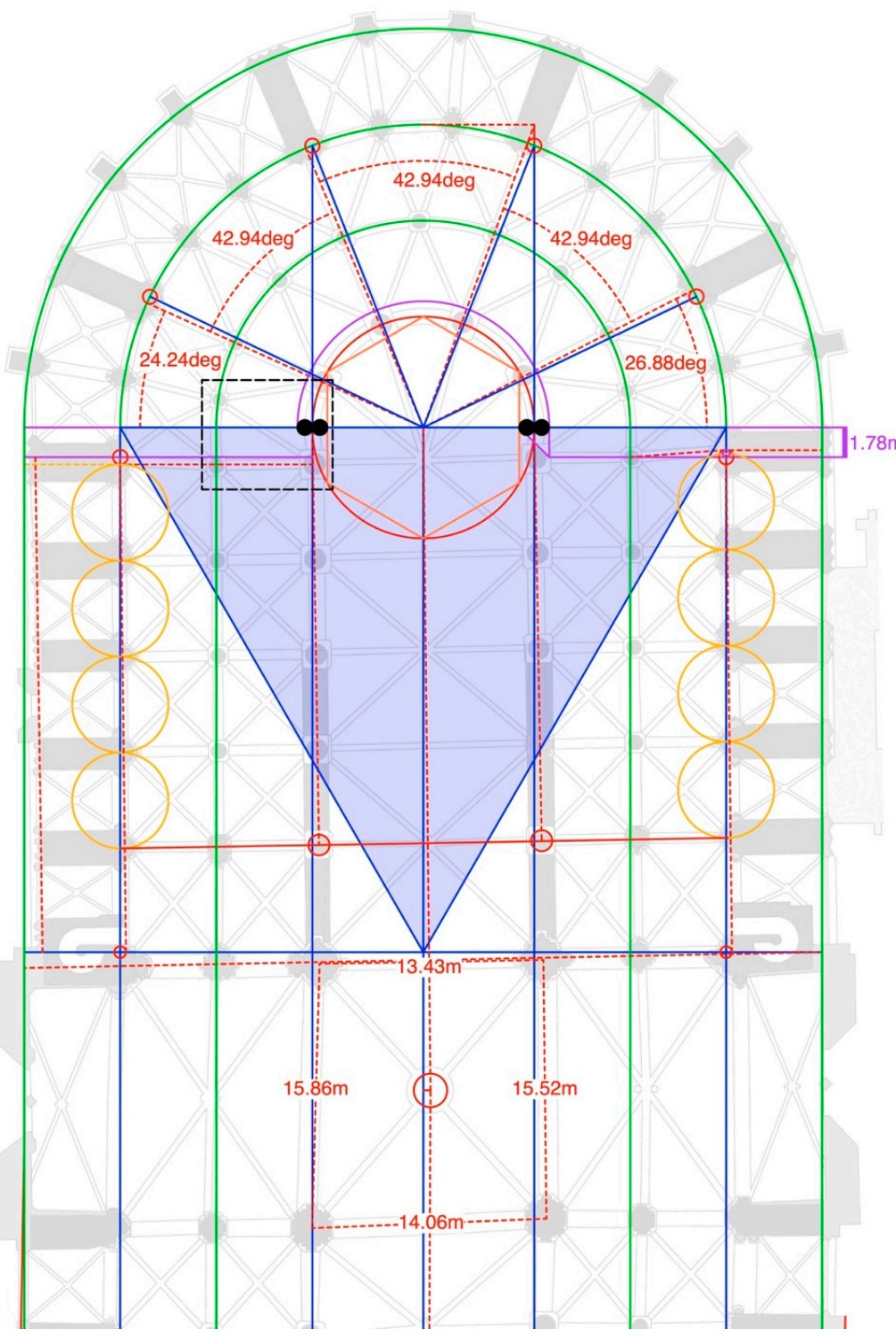

**Figure 9.** Notre-Dame, Paris, geometry of choir plan [geometrical overlays and annotation by the author, from a plan based on a scan by Andrew Tallon and drawn by Laurence Stefanon; © Art Graphique & Patrimoine].

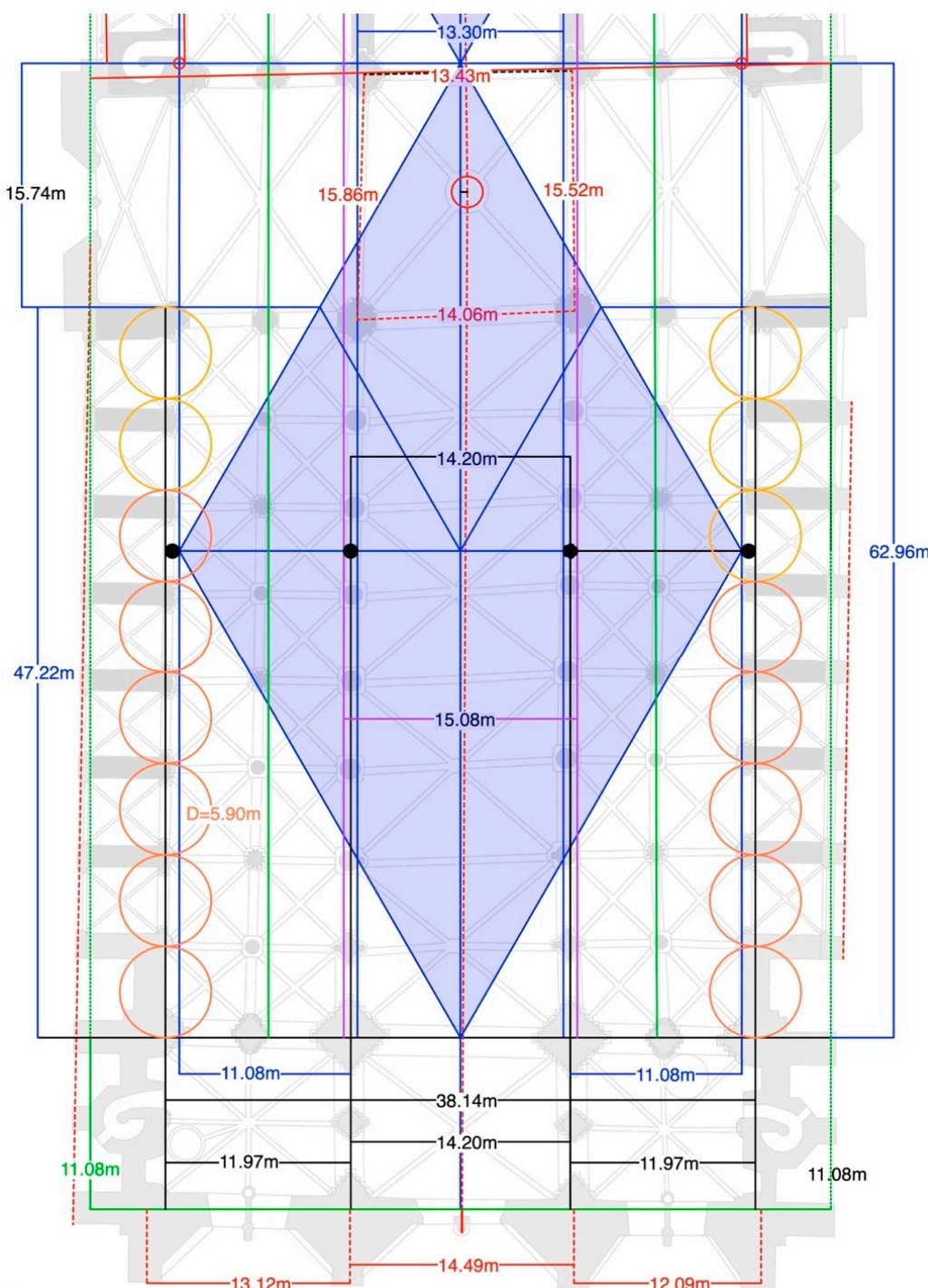

**Figure 10.** Notre-Dame, Paris, geometry of nave plan [geometrical overlays and annotation by the author, from a plan based on a scan by Andrew Tallon and drawn by Laurence Stefanon; © Art Graphique & Patrimoine].

The famous west facade of Notre-Dame, although built in the thirteenth century, similarly seems to have its roots in twelfth-century planning (Figure 11).

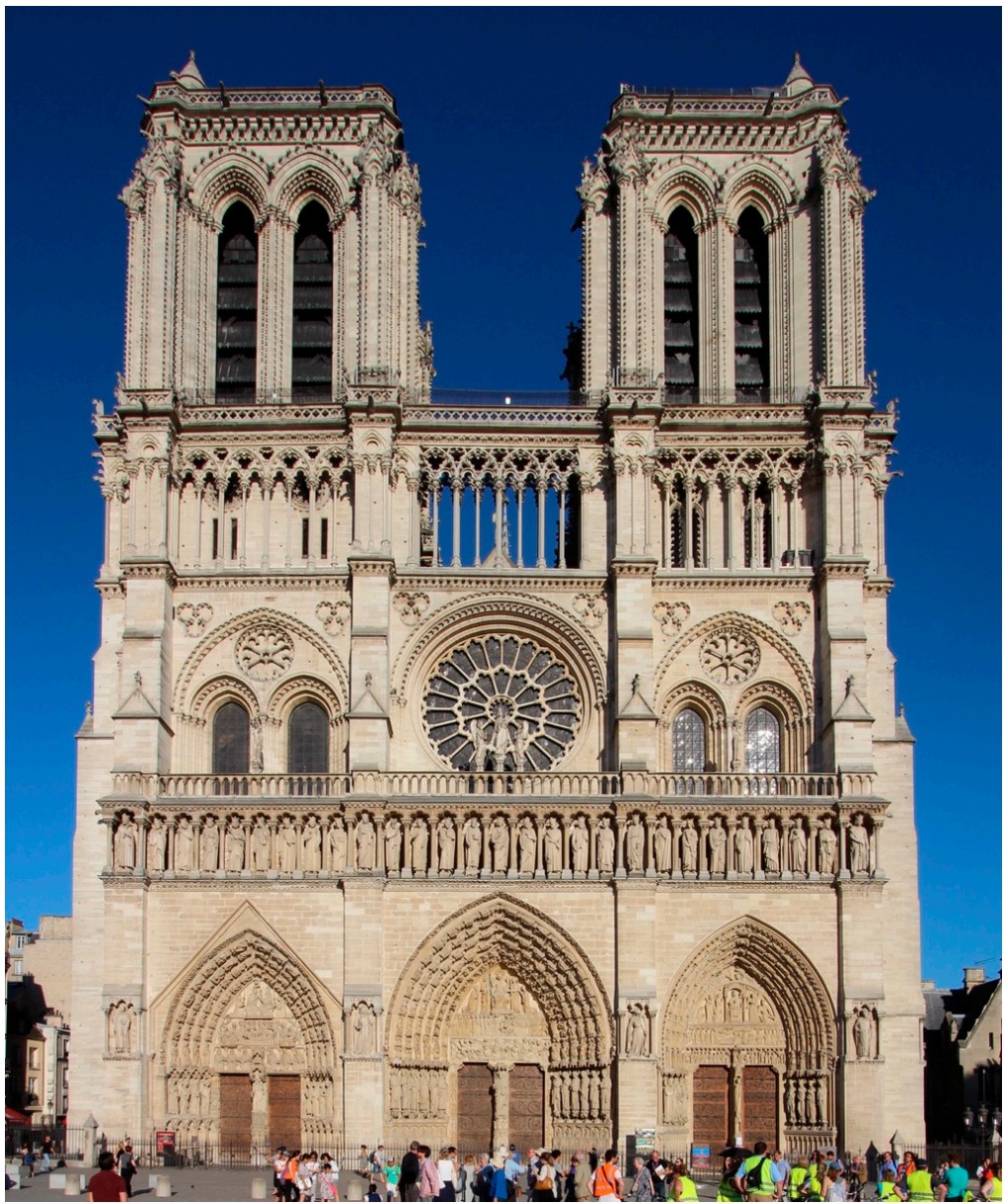

**Figure 11.** Notre-Dame, Paris, west façade [photo by the author].

The present façade nearly fits into the same $\sqrt{3}$ triangle that frames the nave and crossing, as shown in Figure 12. There is good reason to believe, in fact, that the current façade format represents a revision to an original twelfth-century scheme that would have fit exactly within this frame, as part of the first designer's master plan. The current façade block incorporates a tall portal zone, a narrow band of sculptures known as the arcade of kings, and a third story at the level of the rose window. This format does not express the spatial disposition of the cathedral behind it, since the façade portals are taller than the aisles behind them, while the arcade of kings is far shorter than the vaulted gallery story on the corresponding level of the interior.

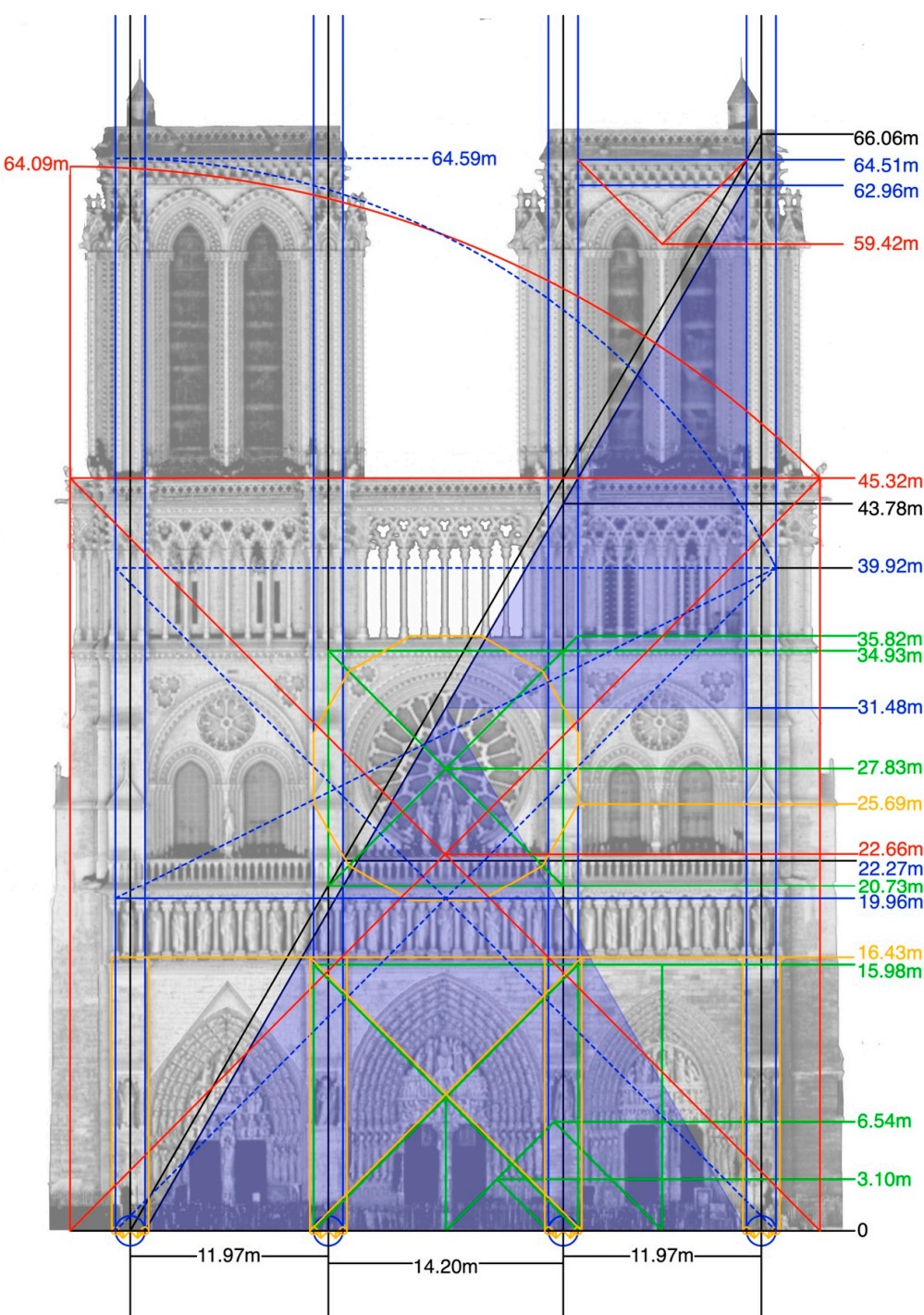

**Figure 12.** Notre-Dame, Paris, geometry of façade, [laser scan data by Andrew Tallon, with proportional rectification, geometrical overlays, and annotation by the author].

If the cathedral's first designer had developed a façade scheme to express the organization of the interior, it would have looked something like Figure 13, with smaller portals than those of the current façade.

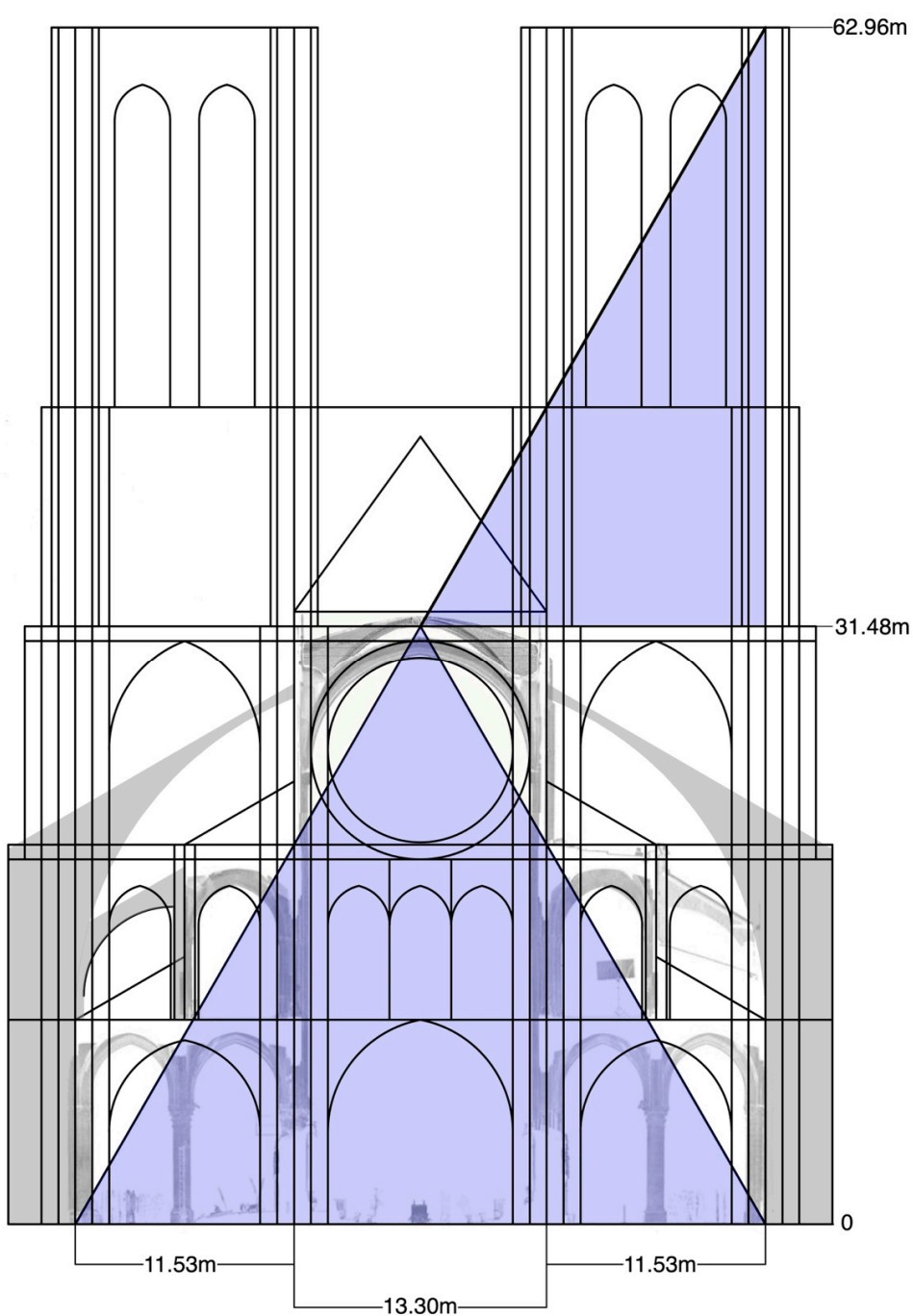

**Figure 13.** Notre-Dame, Paris, hypothetical original façade design [laser scan data by Andrew Tallon, with geometrical overlays and annotation by the author].

Work actually appears to have begun in accord with a façade design like this one, since the southern portal of the current façade incorporates a twelfth-century tympanum clearly designed for a smaller portal. This scheme, moreover, looks very similar to the façade of the collegiate church at Mantes-la-Jolie, a church long recognized as a close cousin of Notre-Dame (Figure 14). It seems likely, in fact, that the Mantes façade was inspired by the original twelfth-century scheme for the façade of Notre-Dame, which could well have been recorded on a parchment drawing.

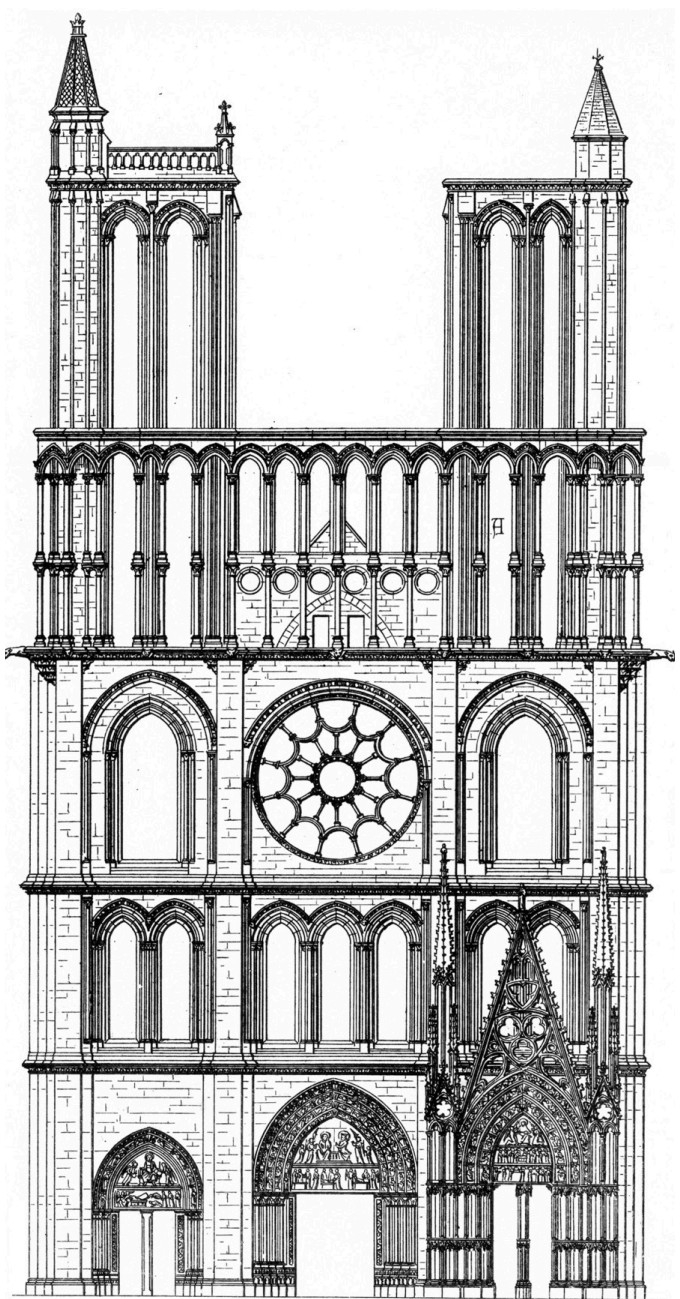

**Figure 14.** Mantes-la-Jolie, collegiate church, façade [from Thomas H. King, *The Studybook of Medieval Architecture and Art* (London: Bell and Daldy, 1858)].

## 6. Conclusions

Examples such as these, to which many more could be added, argue strongly for the use of scaled design drawings in the twelfth century. As noted previously, buildings as complex and sophisticated as Saint-Denis and Notre-Dame likely would have been literally inconceivable without such convenient graphic aids. There is no specific technical or cultural reason, moreover, why such drawings could not have been made a century before the career of Villard de Honnecourt. Indeed, one might expect a breakthrough in design method early in the twelfth century, when the conventions of Gothic design and construction were first coming into focus, rather than in the mid-thirteenth century, when these norms were already well established. The original emergence of the Gothic mode, after all, was surely a more dramatic event than the progressive refinement of bar tracery and linear articulation in Villard's day.

In evaluating the contextual evidence for drawing use in the twelfth century, it is important to note that the use of drawings should not be expected to produce an architectural culture of total control and standardization. In the late Gothic period, for example, the use of drawings facilitated the continuous updating and modification of buildings in progress, as one can see from the drawing collections in centers such as Strasbourg, Ulm, and Vienna. Such ongoing modification remained typical in the Renaissance and Baroque eras, as in the famous cases of Saint Peter's in the Vatican and Saint Paul's in London, despite Leon Battista Alberti's championing of a paradigm in which the architect's pure vision would be recorded in drawings and implemented by builders only after the fact (Trachtenberg 2010). Even in the modern world, the use of mechanical drawings and computer-aided design tools has not resulted in the creation of timeless buildings untouched by modifications made during or shortly after the initial construction process. In evaluating the case for early Gothic drawing use, therefore, the twelfth century should not be held to a standard that even the twenty-first cannot consistently match.

It is certainly unfortunate and inconvenient that no actual small-scale design drawings have survived from the twelfth century. This lack of evidence prevents modern scholars from gaining an intimate perspective into the early Gothic creative process that surviving parchment and paper drawings have provided for the late Gothic era, especially in the Germanic world. This absence of evidence, however, should not be taken as evidence of absence. Although influential scholars from Robert Branner to John James and Günther Binding have argued that such drawings began to be used only in the thirteenth century, the apparent continuity of Gothic design practice in the decades around 1200 argues strongly against this view. The inherent complexity of buildings such as Saint-Denis and Notre-Dame, combined with their linear articulation and the histories of their reception, reinforce the idea that drawing-based planning was already playing an important role in early Gothic architectural culture. When such factors are taken into account, therefore, it becomes clear that the evidence for the use of scaled design drawings in the twelfth century should not be dismissed as paper thin.

**Funding:** This research received no external funding.

**Data Availability Statement:** Not applicable.

**Conflicts of Interest:** The author declares no conflict of interest.

## Notes

[1] On the other drawing collections in the Germanic world, see (Böker et al. 2011) and (Böker et al. 2013). For the much smaller sets of drawings from Spain, Italy, France, and the Low Countries, see respectively: (Alonso Ruiz and Jiménez Martín 2009); (Fernández 2019); (Ascani 1997); (Hamon 2015); and (Hurx 2018).

[2] James Ackerman argues that paper, which became common in Europe only from the fourteenth century onwards, was more suitable for sketching and experimentation than parchment had been, since paper was less expensive. See (Ackerman 2002, p. 294). Parchment drawings were clearly used earlier for the development of design ideas, however, as with the Strasbourg façade drawings discussed below.

[3] For a good review of the scholarship on Villard, see (Barnes 2009). See also (Brooks 2023). For a reassertion of the idea that Villard was an architect, see (Wirth 2015).

[4] Branner cites, for example, an incised pier plan from Saint-Quentin, a series of flying buttress and portal designs traced on the aisle terraces of Clermont, and a rosette from Saint-Albans. See (Branner 1963, pp. 132–33). For discussion of later plaster tracing floors, see (Harvey 1968).

[5] Branner even argued that the richly detailed Strasbourg Plan B was not understood correctly by the builder of the Strasbourg façade. See (Branner 1963, p. 131). The divergences between the drawing and the façade, however, seem instead to involve a careful replanning based on the reformulation of geometrical ideas seen in Plan B. See (Bork 2011, pp. 75–92).

[6] Works that accept Branner's idea that design drawing on parchment became important only after 1200 include: (Bony 1983); (Schöller 1989); (Turnbull 1983); (Pacey 2008); (Bugslag 2008, esp. p. 62); (Davis 2011, esp. pp. 219–23); and (Binding 2014, esp. pp. 36–43). A few scholars, conversely, accept the idea that comprehensive design drawings (and not just full-scale drawings of details) could have been produced before 1200, even if they do not dwell on the subject. Christopher Wilson, for example, notes that "medieval architects must always have relied on fairly detailed drawings to communicate their proposals to their

patrons." See (Wilson 1990, p. 141). Roland Recht, similarly, affirms that "la représentation planimétrique devait exister tout au long du Moyen Âge, en tout cas avant le XIIIe siècle, à des époques où précisément l'essentiel de la structure de l'édifice est contenu dans les indications planimétriques.' See (Recht 1995, p. 27). Most recently, Klaus Jan Philipp has argued that "It can be assumed that Villard was able to refer back to an old tradition. His drawings do not search for a new form of representation..." See (Philipp 2020, p. 21). Meanwhile, the relationship between drawing and other means of design communication has been critically discussed in (Hillson 2020).

7    See, for instance, (James 1977–1982), (James 1989), and most recently (James 2023).

8    See, among others, (Shelby 1981), (Mark 1979), and (Murray 1979).

9    (Binding 2014, pp. 36–43) provides a richly documented survey of primary source citations about the medieval planning and layout process, but several factors weaken its value as an argument against the existence of twelfth-century planning drawings. Many of the cited examples predate the Gothic era, making them irrelevant in this context. Some, moreover, are based on accounts of dubious historicity, such as the story of Gunzo's dream at Cluny. Most fundamentally, though, the idea that many or even most buildings in a given period might be laid out directly on the site without detailed drawings does not imply that all of them were. It is noteworthy, therefore, that Binding cites accounts of drawing-free design practice even from sources written decades after the career of Villard de Honnecourt, including the records of building Wittewierum monastery near Groningen (from 1248) and Thomas Aquinas's more abstract invocations of building practice in his *Summa Theologica* (from 1265). Since such written sources clearly coexisted side by side with the use of parchment drawings in the thirteenth century, one suspects that the same may have been true in the twelfth century, as well. Finally, while written accounts of the building process definitely have their value, any truly satisfying theory of medieval design practice must take into account the forms of the buildings themselves, whose complexity in the Gothic era often challenges the idea of a direct and unmediated translation from the designer's mind to the building site.

10   For a more complete explication of this scheme, see (Bork 2022a, Figures 2 and 3).

11   For a more complete explication of this scheme, see (Bork 2022a, Figure 6). See also https://geometriesofcreation.lib.uiowa.edu/architecture/the-geometry-of-the-choir-plan-in-sugers-saint-denis/ (accessed on 16 October 2023). These analyses are based in large part on the pioneering but unpublished work of Richard Nash Gould, who assisted Sumner Crosby in the surveying of Saint-Denis in the years around 1970.

12   The 27-degree wedges are just slightly smaller than the 27.69-degree wedges that one sees in a regular 13-gon, but they can be constructed far more readily.

13   On Caen, see (Grant 2005, pp. 98–102).

14   See (Bork 2013). See also https://geometriesofcreation.lib.uiowa.edu/2020/06/09/ground-plan-geometries-in-sugers-st-denis-a-prototype-for-altenberg/ (accessed on 16 October 2023).

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
