# Peer review of "Paper Thin? The Evidence for 12th-Century Gothic Design Drawings"

_arts, 2023_

Round 1

Reviewer 1 Report

The work is interesting, focusing mainly on two study cases, although it does not refer to other proportion studies carried out by other researchers.

It would be  also interesting to reference contemporary traces (12th century) found on other buildings.

Author Response

Dear Reviewer,

Thanks very much for your supportive attention to my manuscript.   I am grateful for your endorsement. 

You are right that I have chosen here to concentrate on my own geometrical results rather than citing other proportional studies; I have done so because I have thought more carefully about how my own work relates to the questions of design practice engaged in this essay.  I hope and expect that the publication of my essay may prompt other authors to weigh in with related studies explaining how their work sheds additional light on these questions.

To your point about traces on other buildings, I offer my endnote 11, in which I cite Branner’s mention of the incised pier plan from Saint-Quentin and the rosette from Saint-Albans.  The other comparable examples that I know are all later, which is why I did not include them in this essay.  Again, I hope that further research may bring more examples to scholarly attention.

As I say, I am grateful for your supportive attention.

Sincerely,

Robert Bork

Reviewer 2 Report

In the paper, the question of the existence and use of scaled design drawings with its first verifiable evidence before the 13th century is again addressed. As correctly noted in the paper, the question is still evident, as even the latest survey literature on the medium of architectural drawing (e.g. K.-J. Philipp, Architecture - drawn : from the middle ages to the present, Basel Birkhäuser 2020) gallantly avoids it. The author argues for the existence of such drawings as early as the 12th century. Contrary to the thesis first propagated by Robert Branner (1966) of planning practice without the medium of the work plan without the use of scaled design drawings, which is also represented in a modified form by John James (1977-1982, 1989, 2023), the author refers to the thesis of a continuity of (Gothic) design practice from the 11th century onwards, which presupposes a continuous development and practice of planning by drawing.

The author now convincingly demonstrates the thesis of the indispensable use of scaled 'blueprints' for the buildings of Saint Denis and Notre Dame du Paris with the help of complex graphic-geometric procedures in the ground plan and elevation. In particular, the argument of the transmission of these layout schemes via the medium of drawing is also well comprehensible here. 

The article is clearly structured and argues on the basis of the latest research literature on the subject. However, the introductory passage on the history and state of research omits Günther Binding, an important protagonist of German-language research:  His most recent reservations, expressed in 2014, about the existence of plan drawings before 1200 (G. Binding, Bauwissen im Früh- und Hochmittelalter, Wissensgeschichte der Architektur, Vol. 2, J. Renn, W. Osthues, H. Schlimme, Berlin Edition Open Access, 36-43) and the draft 'in mentem', which he attempts to substantiate above all through numerous early and high medieval textual sources, should once again be critically questioned in the context of the study presented. At the very least, the discrepancy with the contrary descriptions in the sources should be pointed out in passing.

Author Response

Dear Reviewer,

Thanks very much for your supportive attention to my manuscript.   I am grateful for your endorsement, and for your valuable input.

In response to your main suggestion, I have added a passage on page 7 invoking Günther Binding’s 2014 essay.  I know Binding’s earlier work, and I think highly of it, but I did not know this more recent reference, and I am glad that you brought it to my attention.  As I explain in a newly added footnote, Binding “provides a richly documented survey of primary source citations about the medieval planning and layout process, but several factors weaken its value as an argument against the existence of twelfth-century planning drawings.  Many of the cited examples predate the Gothic era, making them irrelevant in this context.  Some, moreover, are based on accounts of dubious historicity, such as the story of Gunzo’s dream at Cluny. Most fundamentally, though, the idea that many or even most buildings in a given period might be laid out directly on the site without detailed drawings does not imply that all of them were.  It is noteworthy, therefore, that Binding cites accounts of drawing-free design practice even from sources written decades after the career of Villard de Honnecourt, including the records of building Wittewierum monastery near Groningen (from 1248) and Thomas Aquinas’s more abstract invocations of building practice in his Summa Theologica (from 1265).  Since such written sources clearly coexisted side by side with the use of parchment drawings in the thirteenth century, one suspects that the same may have been true in the twelfth century, as well.  Finally, while written accounts of the building process definitely have their value, any truly satisfying theory of medieval design practice must take into account the forms of the buildings themselves, whose complexity in the Gothic era often challenges the idea of a direct and unmediated translation from the designer’s mind to the building site.”   I hope that these points will make sense from your perspective, and that they will suffice to explain my position. 

I have also added a brief citation to the work of Klaus Jan Philip in my endnote 19.  I say “Most recently, Klaus Jan Philipp has argued that “It can be assumed that Villard was able to refer back to an old tradition. His drawings do not search for a new form of representation…” See (Philipp 2020, p. 21).”  Again, I thank you for alerting me to this reference, since it was nice to have another 21st-century author sympathetic to pre-Villard Gothic drawings. 

As I say, I am truly grateful for your supportive attention.

Sincerely,

Robert Bork

Reviewer 3 Report

This paper is an important contribution to architectural studies.  It focuses on architectural drawings and their role in the Gothic Europe. Most discussions about architectural drawings and, by extension, about architectural design in medieval times start with Villard de Honnecourt. His 13th-century sketchbook with about 250 different illustrations includes architectural drawings, and scholars use them as evidence of the beginning of the use of architectural drawings in Western Europe. This paper convincingly challenges this assumption on the singular role of Villard de Honnecourt, highlights the teamwork inherent to architectural practice, and furthers the argument on the role of architectural drawings for architectural design.

The author logically argues that architectural drawings must have been used in the twelfth century, and likely earlier, despite the absence of surviving drawings. The architectural accomplishments are evidence in their own right. The earliest large-scale Gothic buildings of the twelfth century display a high level of geometrical sophistication and overall coherence that cannot be achieved without architectural design and proficiency. The author reconstructs the sophisticated geometric schemas applied to the twelfth-century phases of Saint-Denis Abbey and Notre Dame de Paris. Preserved scaled architectural drawings are dated to the 13th and 14th centuries, but as the author wittingly remarks, “the absence of evidence (for their earlier use) should not be understood as evidence of absence.”

Moreover, the corpus of some 650 architectural drawings from the Gothic period (mostly floor plans and elevations), most of them (more than 400) in Vienna, allows the author to analyze and underscore the role of architectural drawings in communicating and transferring architectural knowledge. The author opens a much-needed discussion on architectural networks in a wider map of Western Europe. The mature understanding of architectural drawings in the transfer and development of architectural knowledge and practices is also evident in the author’s explanation that they were not used to make exact replicas of Gothic churches in various locations but as generative tools in devising new solutions. Hence, the paper makes a solid argument for drawing-based architectural design in medieval Europe, earlier than previously claimed by some authorities. Moreover, the paper presents important questions when discussing architectural design and practices in the Middle Ages, where we often lack an understanding of architecture as a discipline based on surviving textual sources.

I highly recommend this paper for publication.

With best wishes to the author and editors of the journal for continual success in advancing and promoting high-quality scholarly work.

Author Response

Dear Reviewer,

Thanks so much for your generous and even effusive comments about my manuscript.  As you can imagine, your positive feedback made my day.  I am pleased to have had the chance to prepare this article, and I hope that it will come to be seen as an important contribution, as you kindly suggested that it might.  I am truly grateful for your support and engagement!

Best wishes,

Robert Bork